# Prevalence of HIV-1 Natural Polymorphisms and Integrase-Resistance-Associated Mutations in African Children

**DOI:** 10.3390/v15020546

**Published:** 2023-02-16

**Authors:** Djeneba B. Fofana, Houdou Diarra, Ibrahima Guindo, Mahamadou K. Savadogo, Marceline d’Almeida, Fatoumata I. Diallo, Aliou Baldé, Cathia Soulié, Amadou Kone, Anne-Geneviève Marcelin, Almoustapha I. Maiga, Sidonie Lambert-Niclot, Mamoudou Maiga, Sally McFall, Claudia A. Hawkins, Robert L. Murphy, Mariam Sylla, Christine Katlama, Jane L. Holl, Vincent Calvez, Laurence Morand-Joubert

**Affiliations:** 1Faculty of Medicine, University of Sciences, Techniques and Technologies of Bamako (USTTB), Bamako E 423, Mali; 2Sorbonne Université, INSERM, Institut Pierre Louis d’Epidémiologie et de Santé Publique (iPLESP), F-75012 Paris, France; 3Department of Virology, Assistance Publique-Hôpitaux de Paris (AP-HP), Saint-Antoine Hospital, F-75012 Paris, France; 4Centre d’Ecoute, de Soins, d’Animation et de Conseils (CESAC), Bamako E 2561, Mali; 5Département Mère Enfant, Faculté Des Sciences De La Santé, Université Abomey-Calavi, CNHU—HKM, Cotonou 229, Benin; 6Department of Virology, Assistance Publique-HÔpitaux de Paris (AP-HP), Pitié Salpêtrière Hospital, F-75013 Paris, France; 7Institute for Global Health, Northwestern University, Chicago, IL 60208, USA; 8Service des Maladies Infectieuses, Hôpital Pitié-Salpêtrière APHP, F-75013 Paris, France; 9Biological Sciences Division, University of Chicago, Chicago, IL 60637, USA

**Keywords:** HIV-1, polymorphism, integrase, children, resistance, Africa

## Abstract

Integrase inhibitors (INIs) are a potent option for HIV treatment. Limited data exist on INI resistance in West Africa, particularly in children living with HIV/AIDS. We determined the prevalence of integrase gene polymorphisms and the frequency of naturally occurring amino acid (aa) substitutions at positions associated with INI resistance. Dried blood spot (DBS) samples were obtained from one hundred and seven (107) HIV-1-infected children aged less than 15 years old in two West African countries, Benin and Mali. All children were naïve to INI treatment, 56 were naïve to anti-retroviral therapy (ART), and 51 had received ART. Genetic sequencing of HIV integrase was successful in 75 samples. The aa changes at integrase positions associated with INI resistance were examined according to the Stanford HIV Genotypic Resistance database. The median ages were 2.6 and 10 years for ART-naïve and -treated children, respectively. The most common subtypes observed were CRF02_AG (74.7%) followed by CRF06_cpx (20%). No major INI-resistance mutations at positions 66, 92, 121, 143, 147, 148, 155, and 263 were detected. The most prevalent INI accessory resistance mutations were: L74I/M (14/75, 18.6%) followed by E157Q (8/75, 10.6%), G163E/N/T/Q (5/75, 6.6%), Q95A/H/P (2/75, 2.6%), and T97A (4/75, 5.3%). Other substitutions observed were M50I/L/P, H51E/P/S/Q, I72V, T112V, V201I, and T206S. Polymorphisms at positions which may influence the genetic barrier and/or drive the selection of specific INI-resistance pathways were detected. However, no transmitted drug resistance (TDR) to INI was detected among samples of INI-naïve patients. These findings support the use of this treatment class for children with HIV-1, particularly in West Africa.

## 1. Introduction

Major efforts have been directed towards the development of drugs targeting HIV-1 integrase, a crucial enzyme involved in HIV-1 replication [1]. Integrase-inhibitor (INI)-based drugs block the integrase strand transfer, that is, the binding of the INI to the integrase enzyme, thereby preventing the integration of the viral genome into the host DNA [2]. First-generation INIs, raltegravir (RAL) and elvitegravir (EVG), were approved by the United States Food and Drug Administration (US FDA) for clinical use in 2007 and 2012, respectively, with the second-generation INIs dolutegravir (DTG), bictegravir (BIC), and cabotegravir (CAB) being approved in 2012, 2018, and 2021, respectively [3]. INI-based regimens are presently among the most commonly used and recommended first-line antiretroviral therapies (ARTs) for HIV-1 infection, in both treatment-experienced and ART-naïve patients, due to their high potency, effectiveness, safety, low toxicity, and tolerability [1,4,5,6]. The combined use of INI-based drugs with other early therapies, such as nucleoside reverse-transcriptase inhibitors (NRTIs), the non-nucleoside reverse-transcriptase inhibitor (NNRTI) Etravirine, and the protease inhibitor (PI) Darunavir, have resulted in high levels of viral-load suppression in HIV-infected patients [4,6]. However, the long-term efficacy of these drugs has been restricted by the emergence of resistance-associated mutations [7,8].

An estimated 1.8 million children aged 0–14 were living with HIV at the end of 2019, and 150.000 children were newly infected. The sub-Saharan African region remains the most affected. Benin and Mali, two West African countries, have experienced similar HIV-1 epidemics, with prevalence rates of around 1% (UNAIDS report, 2021); the most prominent strain circulating within these countries is CRF02_AG [9]. HIV-positive children should be started on antiretroviral drugs (ARVs) immediately, although in 2018 almost half of all children living with HIV were not taking ARVs.

A multi-country analysis of HIV drug resistance in sub-Saharan African countries showed that 53.0% and 8.8% of newly diagnosed infants had resistance to one or more non-nucleoside reverse-transcriptase inhibitors (NNRTIs) and nucleoside reverse-transcriptase inhibitors (NRTIs), respectively, before treatment initiation [10]. The World Health Organization WHO) recommended dolutegravir-based ART as the preferred first- and second-line treatment for adults and children with HIV-1 infection [11]. In Benin and Mali, DTG and RAL are the only two INIs accessible since 2018. 

The ODYSSEY trial, a randomized study that included African countries, showed evidence of the superior efficacy of dolutegravir-based ART, compared with standard care in children and adolescents, as first-line and second-line ART [12].

Since the implementation of INI regimens, surveillance of INI-selected mutations has gained in importance [13]. Several INI-resistance-associated mutations (RAMs) have been reported [7]. Three groups (major, accessory, and other mutations) have been classified in the Stanford HIV Drug Resistance Database (http://hivdb.stanford.edu, accessed on 2 January 2023) based on their associations with reduced drug susceptibility: (1) major mutations represent non-polymorphic mutations that, by themselves, reduce susceptibility to one or more INIs; (2) accessory mutations represent polymorphic or non-polymorphic mutations that can reduce INI susceptibility in combination with major RAMs; (3) other mutations are polymorphic or non-polymorphic mutations that can be selected under INI-based therapy and which may or may not potentially reduce susceptibility [1].

In developed counties, INI RAMs have been well studied but there are few data for Africa, particularly in children. The aim of this study was to characterize the prevalence of HIV-1 natural polymorphisms and RAMs in INI-naïve, HIV-1-infected children in two West African countries.

## 2. Materials and Methods

### 2.1. Patients and Sample Collection

Dried blood spots (DBSs) were obtained from 107 HIV-1-infected children aged less than 15 years old living in two West African countries, Mali and Benin. The DBSs were collected from 51 ART-treated children with virological failure, defined as having at least one viral load (VL) >1000 copies/mL, during a study conducted in Benin between 2015 and 2016 [14] and from 56 ART-naïve children, collected between 2018 and 2020 during early diagnosis testing and follow-up in Mali. All children were INI-naïve. Whatman 903 filter paper was used to collect the DBSs, with 50 uL of blood being dropped into each concentric circle. The DBSs were dried at ambient temperature and then sent to the Virology Laboratory at the Saint-Antoine Hospital (Paris, France) for sequencing. 

### 2.2. Genotyping and Analysis of Resistance-Associated Mutations

HIV-1 RNA was extracted from the 107 DBSs using EasyMag technology as per the manufacturer’s instructions. Target integrase regions of the HIV-1 were subsequently amplified by PCR after reverse-transcription production of complementary DNA (cDNA) and sequenced using the Sanger method. Genotypic-resistance testing was conducted on the integrase gene’s target regions of HIV-1 using the ANRS ((French) National Agency for Research on AIDS and Viral Hepatitis) method (http://www.hivfrenchresistance.org/, accessed on 2 January 2023). 

The primers used were: outer primers: INPS1: 5′-TAG TAG CCA GCT GTG ATA AAT GTC-3′ and INPR8: 5′-TTC CAT GTT CTA ATC CTC ATC CTG-3′; inner and sequencing primers: INPS3: 5′-GAA GCC ATG CAT GGA CAA G-3′ and INPR9: 5′-ATC CTC ATC CTG TCT ACT TGC C-3′. The PCR conditions were as follows: RT-PCR step, using the RT-PCR kit Access (PROMEGA): 45′ at 45 °C, 2′ at 95 °C, then: (30″ at 95 °C; 30″ at 50 °C; 1′ at 68 °C) × 40 cycles. Nested PCR step, using Ampli Taq Polymerase with Gene Amp (Roche): 5′ at 94 °C, then (30″ at 94 °C; 30″ at 55 °C (50 °C); 1′ at 72 °C) × 45 cycles 7′ at 72 °C.

Seventy-five (75) of the 107 samples were successfully sequenced, assembled, and aligned using Sescap and SmartGene HIV Software (Innovation Park, Lausanne, Switzerland). The other 32 samples could not be amplified. The INI sequences, successfully analyzed from 75 DBS samples, were from 31 ART-treated children with virological failure (20 from Benin and 11 from Mali) and 44 ART-naïve HIV-1-infected Malian children. Positions associated with INI resistance were analyzed, and the interpretation of INI resistance was performed following the Stanford HIV resistance algorithm, version 7.0. The HIV-1 subtypes and CRFs were determined according to the Stanford University HIV Drug Resistance Database HIVdb Program, latest version (https://hivdb.stanford.edu/hivdb, accessed on 2 January 2023), with a single integrase gene amplification. Regarding INI mutations, all mutations reported in the IAS and Stanford lists were considered; in particular, the major INI RAMs from the latest version of Stanford’s list, including T66A/I/K, E92Q, G118R, E138K/A/T, G140S/A/C/R, Y143R/C/H, S147G, Q148H/R/K, N155H, and R263K (https://hivdb.stanford.edu/, accessed on 2 January 2023). Some uncommon nonpolymorphic accessory mutations, such as M50I/L/P, H51Y, and F121Y, alone, do not reduce INI susceptibility [15]. We also analyzed 19 natural polymorphisms (I72V, L74I/M, T97A, T112I, A128T, E138K, Q148H, V151I, S153Y, S153A, M154I, N155H, K156N, E157Q, G163R, V165I, V201I, I203M, and T206S) [16]. HIV-1 strains were defined as resistant if they carried at least one major transmitted drug-resistance (TDR) mutation or natural polymorphism. The overall prevalence was defined as the percentage of HIV-1-infected patients with any strain with a TDR mutation or natural polymorphism.

All sequences were submitted to GenBank and registered under accession numbers OQ435656-OQ435729. 

### 2.3. Data Analysis

The participant characteristics and the proportions of resistance or polymorphism mutations were compared between the naïve and treated patients. Fisher’s test was used to compare categorical variables, and the Wilcoxon test was used to compare continuous variables. All statistical analyses were computed using STATA, version 11 (Stata Corporation, College Station, TX, USA). All statistical tests were two-tailed, with α = 0.05.

## 3. Results

### 3.1. Characteristics of Study Subjects

The characteristics of the patients providing the 75 samples that were successfully sequenced are shown in Table 1. At enrollment, 90.3% of the treated children were receiving non-nucleoside-reverse-transcriptase-inhibitor (NNRTI)-based regimens, and only 9.7% received both nucleoside-reverse-transcriptase inhibitor (NRTI)- and PI-based regimens. INI-resistance mutations in sequenced samples are shown in Table 2.

### 3.2. INI-Resistance-Associated Mutations and Natural Polymorphism Patterns

No residue substitutions of major INI RAMs at codons 66, 92, 121, 143, 147, 148, 155, or 263 were observed in the 75 sequences. Of the 75 sequences, 25/75, 33%, harbored at least one INI accessory polymorphism mutation (L74I/M, Q95A/H/P, T97A, G118V, E138Q, S153F, E157Q, G163E/N/T/Q, or S230N) and 48 (64%) harbored at least one polymorphism mutation, described as having no impact on INI sensitivity [15,16].

Among the detected INI accessory mutations, L74I/M was the most prevalent (14/75, 18%), followed by E157Q (8/75, 10.6%), G163EN/T/Q (5/75, 6.6%), Q95A/H/P (2/75, 2.6%), and T97A (4/75, 5.3%). The lowest prevalence of residue substitutions was found at codons G118V, S153F, and S230N, all with the same prevalence (1.3%), representing only 1 of the 75 samples. The prevalence of E138Q was 2.6% (2/75) (Figure 1). Some natural polymorphism patterns (no known impact on INI sensitivity) were also observed (Figure 2). The highest prevalence in this group was observed at codon T206S (74, 7%), followed by T112V (62.6%), V201I (61/75, 81.3%), and V72I (26/75, 34.6%). Other mutations, H51E/P/S/Q and M50I/L/P, were found to have prevalences of 28% and 26%, respectively. The prevalence of INI accessory polymorphism mutations at the described positions was higher in samples with ART than those without ART, particularly the E157Q, E138Q, T97A, and Q95A/H/P mutations (Figure 1), whereas, the presence of polymorphisms described as having no known impact on INI sensitivity was higher in ART-naïve patient samples (Figure 2).

## 4. Discussion

This first report from Benin and Mali found that no major INI RAMs were identified, consistent with prior studies [7,14,18], further supporting evidence that primary INI RAMs are rare. This finding is congruent with studies from the United States [19] and Europe [20], where INI drug-resistance mutations (DRMs) have rarely been identified in ART-naïve patients. The absence of major INI DRMs among INI-naïve patients in our study was also consistent with studies from other African countries, such as Zimbabwe, Uganda, Morocco, and Cameroon [21,22,23,24]. However, a multi-country study from SSA, including Kenya, South Africa, Uganda, and Zambia, found a 2% prevalence of INI RAMs using Illumina next-generation sequencing [25]. This difference can be explained by the fact that our analysis was based on the Sanger sequencing method, which might have underestimated the prevalence of INI RAMs. The Sanger method does not detect drug-resistance minority variants below 20% of the virus population. The most frequently reported major INI RAMs include codons T66A/I/K, E92G/Q, F121Y, Y143C/H/R/S, S147G, Q148H/R/K, N155H, and R263K [1,13,26], while accessory RAMs and other mutations have been largely reported in recent studies from SSA [21,22,23,24,25,27]. The prevalence of both major and accessory mutations in the ART-naïve population is very low (0.5%) and likely results from cases of transmitted INI resistance and the cumulative natural occurrence of mutations without selective drug pressure [4,26,28]. Nine accessory INI-resistance mutations were found in this study, among which L74I/M, T97A, and E157Q, were identified most frequently. These mutations were observed in Cameroon and Ethiopia [24,27], and L74M/I and M50I have been reported in the untreated population. Although many reports have shown that accessory INI-resistance mutations (including naturally occurring polymorphisms) do not compromise the effectiveness of INI-based regimens [29,30], other studies have revealed that several mutations can act synergistically or additively to decrease drug susceptibility [31,32]; for instance, the combination of L74M and G163R associated with T79A, has been reported by Abram et al. as further reducing susceptibility to RAL and/or EVG in the absence of primary INI-resistance mutations. Therefore, the accessory INI-resistance mutations found in this study should be closely monitored, as they may develop synergistic combinations. In addition, some accessory INI RAMs may also pre-exist in treatment-naïve patients. Although prevalence was low, a prior study reported that prevalence can increase over time and emerge in combination with nonpolymorphic INI-resistance mutations [26]. The subtypes found in our study were non-B subtypes, notably, CRF02_AG and CRF06_cpx, which are prevalent in West Africa. L74M tends to be more prevalent in non-B subtypes [1], explaining its high prevalence in our sequenced samples. The second most common mutation was E157Q in those sequences, a polymorphic mutation that was weakly selected in RAL-treated patients and in vitro with EVG. E157Q was the most common polymorphic accessory mutation detected in an Ethiopian analysis, and natural polymorphisms were present in 1–10% of untreated individuals, depending on the subtype [27]. It has been shown that E157Q decreases RAL susceptibility five-fold and EVG two-fold [33]. In addition, this mutation may increase DTG resistance because it may be a compensatory mutation that partially restores the enzymatic activity and infectivity lost with the R263K mutation [34]. However, no samples contained R263K in this study. This mutation was not detected in any non-B specimens in a Canadian study [1]. 

The mutation T97A is a naturally occurring and lowly prevalent integrase polymorphism that emerges more frequently among patients carrying non-B, as in our setting, compared to those carrying HIV subtype B [7,18,29,33,35,36]. The prevalence of T97A (5.6%) was high compared to prior studies [7,35,37]. T97A is associated with low-level decreased susceptibility to EVG and/or RAL [38,39] and has been detected in patients with virological failure and receiving INI-based therapy [7,40,41,42]. Therefore, the presence of T97A in this study suggests the need to evaluate the efficacy of INI-based treatment in HIV-infected patients. INI-associated polymorphic substitution of E157Q or L74I was identified during the IMPAACT P1093 study which evaluated the safety, tolerability, efficacy, and pharmacokinetics of dolutegravir in combination with an optimized background regimen for the treatment of HIV-1 in infants, children, and adolescents aged 4 weeks to 18 years in age-defined cohorts [43].

Some substitutions with no known effect on INI susceptibility were also identified, (V72I, T112V, V201I, T206S, Y227F, and S57N/G), with prevalences from 5% to 75%. Such frequencies can be attributed to natural genetic variability, frequently observed in the integrase gene [35,44]. The oscillation of the frequency of the unknown-effect polymorphism mutations may result from genetic drift [45] since HIV-1 is subject to genetic variability. Further investigations are required to elucidate the association of these unknown mutations with INI-based regimens.

We have reported a higher prevalence of some accessory INI-resistance mutations in ART-treated samples compared to untreated samples. Indeed, 15% of naïve patients and 11% of treated patients harbored at least one INI accessory polymorphism mutation at described positions. These prevalences were statically significant, with 38% and 17% (*p* = 0.0035) prevalences observed for other polymorphism mutations in naïve and treated patients, respectively. The presence of these mutations in ART-treated though INI-naïve patients is conceivable, since reports have shown that integrase mutations increase their prevalence in ART-treated compared to ART-naïve patients, suggesting that some NNRTIs, NRTIs, or PIs could interact (directly or indirectly) with the integrase inducing or selecting INI secondary mutations [29]. However, the presence of accessory INI-resistance mutations in ART-naïve patients may also result from transmitted drug mutation, mostly acquired by children via mother-to-child transmission, as demonstrated with NNRTI TDR in Africa [7,46]. Although naturally occurring integrase polymorphisms generally have little or no clinical effect on INI susceptibility [29], they are of concern when they evolve into major resistance mutations, as this could facilitate the viral evolution of resistance under INI pressure [35]. Indeed, INI-based therapies, according to WHO guidelines, are currently used, even in children. INIs, such as raltegravir and dolutegravir, are increasingly used in both ART-naïve and -treated patients in Africa. In addition, in a recent study, the authors reported that the major INI-associated mutation represents a public-health concern whether natural or transmitted, since it may affect treatment efficacy and negatively affect prognosis [13]. We also recommend future studies to consider the mutations in the nef 3′-PPT region, which suggest an alternative pathway to resistance to DTG and other INIs [24].

## 5. Conclusions

Currently, INI-based treatments are associated with excellent clinical results in patients, leading to a healthier lifestyle and reducing the risk of viral transmission. No TDR to INI was detected among the sequences from INI-naïve child patients. These data support using this class of ARTs in West Africa, including in the pediatric population. However, continued surveillance of INI RAMs is recommended to prevent the emergence of resistance to this class of drugs.

## Figures and Tables

**Figure 1 viruses-15-00546-f001:**
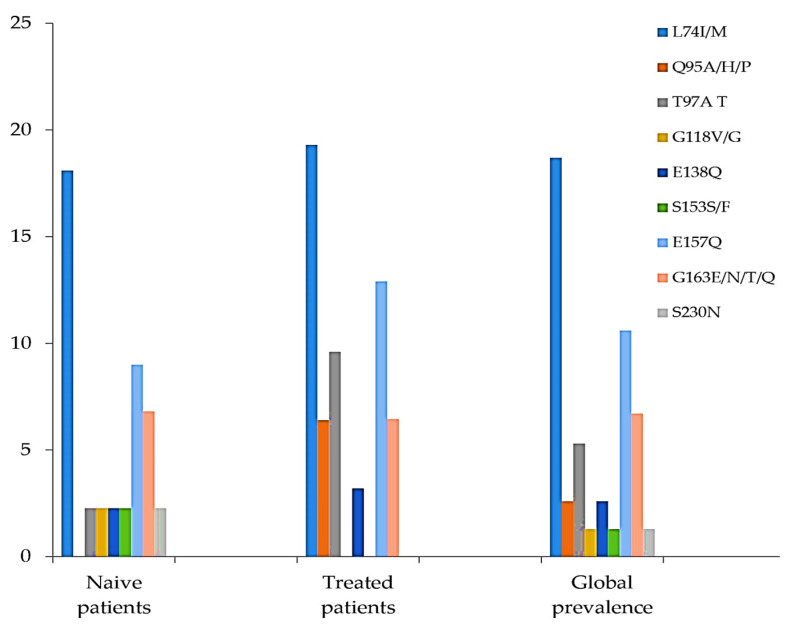
Prevalence of INI accessory polymorphism mutations at described positions.

**Figure 2 viruses-15-00546-f002:**
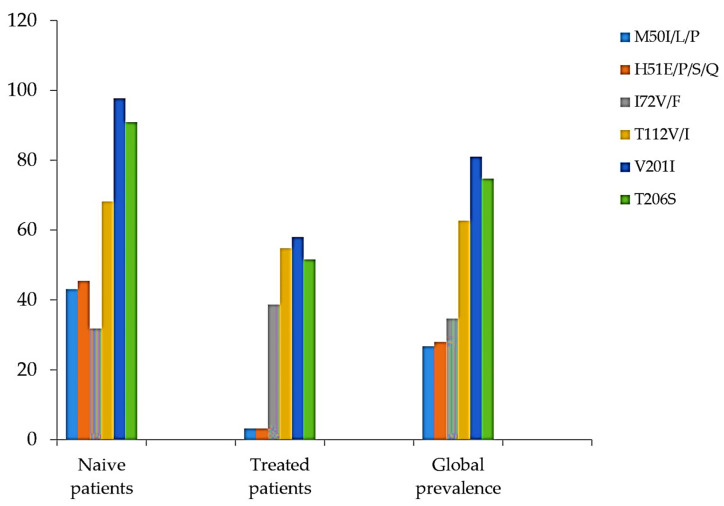
Presence of polymorphism mutations described as having no known impact on INI sensitivity (Adapted from [17]).

**Table 1 viruses-15-00546-t001:** Characteristics of Patients Providing the Sequenced Samples.

Characteristics	ART-Naïve (*n* = 44)	ART-Treated (*n* = 31)	Total	*p*-Value
Gender, *n* (%)				
Male	11 (25.0)	24 (77.4)	35/75 (46.6%)	<0.0001 **
Female	21 (47.7)	7 (22.6)	28/75 (37.4%)	
Not available	12 (27.3)	−0 (0.0)	12/75 (16%)	
Age, year, median (IQR)	2.6 (1.5–5.0)	10.0 (6.0–13.0)		0.0005
CD4 cell/mm^3^, median (IQR)	538	362		0.0659
	(341–1003)	(170–607)		
RNA HIV-1 copies/mL, median (IQR) *	ND	54,000 (5543–170,000)		
Duration of treatment, years, median (IQR) *	NA	5 (3–7)		
Treatment at enrollment *, *n* (%)				
3TC + (ZDV or ABC) + NVP or EFV	NA	28 (90.3)		
TDF + (3TC or FTC or ABC) + LPV/r	NA	3 (9.7)		
Subtypes, *n* (%)				
CRF02_AG	34	22 (71.0)	56/75 (74.7%)	0.5957 ***
CRF06_cpx	9	6 (19.3)	15/75 (20.0%)	
Other	0	3 (9.7)	3/75 (5.3%)	
INI accessory polymorphism mutations at described positions, *n* (%) ****	15 (34.1)	11 (35.5)		1.0000
Other polymorphism mutations, *n* (%) ****	38 (86.4)	17 (54.8)		0.0035

* Treated children only. Abbreviations: 3TC: Lamivudine, ZDV: Zidovidune, ABC: Abacavir, NVP: Nevirapine, EFV: Efavirenz, FTC: Emtricitabine, TDF: Tenofovir, LPV/r: Lopinavir/ritonavir, ND: No data available, NA: Not applicable. ** *p*-values were obtained via Fisher’s test for categorical variables and Mann–Whitney tests for continue variables. *** *p*-value was obtained via Fisher’s test, comparing CRF02_AG versus others (CRF06_cpx+other). **** Number of sequences with at least one mutation.

**Table 2 viruses-15-00546-t002:** INI-Resistance Mutations in Sequenced Samples.

Mutations	ART-Naïve	ART-Treated	Number Totals of Sequences with at Least One Mutation
INI accessory polymorphism mutations at described positions, *n* (%)			
L74I/M	8/44 (18.1)	6/31 (19.3)	14/75 (18.7)
Q95A/H/P	0	2/31 (6.4)	2/75 (2.6)
T97A/T	1/44 (2.3)	3/31 (9.6)	4/75 (5.3)
G118V/G	1/44 (2.3)	0	1/75 (1.3)
E138Q	1/44 (2.3)	1/31(3.2)	2/75 (2.6)
S153F	1/44 (2.3)	0	1/75 (1.3)
E157Q	4/44 (9)	4/31 (12.9)	8/75 (10.6)
G163E/N/T/Q	3/44 (6.8)	2/31 (6.4)	5/75 (6.7)
S230N	1/44 (2.3)	0	1/75 (1.3)
Other polymorphism mutations, *n* (%)			
M50I/L/P	19/44 (43.1)	1/31 (3.2)	20/75 (26.7)
H51E/P/S/Q	20/44 (45.4)	1/31 (3.2)	21/75 (28)
I72V/F	14/44 (31.8)	12/31 (38.7)	26/75 (34.7)
T112V/I	30/44 (68.1)	17/31 (54.8)	47/75 (62.7)
V201I	43/44 (97.7)	18/31 (58.0)	61/75 (81)
T206S	40/44 (90.9)	16/31 (51.6)	56/75 (74.7)

## Data Availability

Data available upon request from the corresponding author.

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
