# Peer review of "Prevalence of HIV-1 Natural Polymorphisms and Integrase-Resistance-Associated Mutations in African Children"

_viruses, 2023, doi:10.3390/v15020546_

Round 1

Reviewer 1 Report

The manuscript "Prevalence of HIV-1 Natural Polymorphisms and Integrase Resistance Associated Mutations in African Children" by Fofana et al., describes the integrate inhibitor resistance mutation profile in child leaving with HIV from two African countries. As none of the patients were exposed to integrase strand transfer inhibitors, the outcome of the study is quite predictable and descriptive in nature.

The manuscript should address the followings,

1. As the study subjects were all Childs the the transmission route of HIV should be described, and if it is a case of mother to child transmission, the HIV status of their mothers should be discussed......means if the mothers were exposed to integrase inhibitors or not, their viral suppression history etc, to understand the vertical transfer of the DRMs. 

2. Recently mutations in the 3'-PPT (polypurine tract) is also associated with effectiveness of the integrase Strand-transfer inhibitors (INSTIs). So, it is recommended to screen this region of the viral genome in this study population.

3. The sequences of the viral genome generated in the study, should be submitted in gene bank and the corresponding accession numbers be included in the manuscript.

4. All studies including the adults or Childs, that look at the viral genome sequence of integrase gene from West Africa should be included in the discussion while comparing the findings of the present study.

5. The  method used to amplify the viral genome should be described in details along with the primer sequences used in the study to help in replication of the study in future.

6. The genotype of all the samples should be included which described as others in the table 1.

7. In table 1, authors describe 44 samples as naive and 31 treated. But then they include ART details of 44 samples...this needs to be double checked and rectified.

In table 1, the vl and CD4 data should be segregated among the drug naive patients, treated with virology failure and treated with viral suppression.

8. In figure 1 and 2, the global prevalence data should be segregated as naive and treated group, this will help to better compare the global data vs data from the present study.

9. In line 147, for t206s, the % is not correct.

10. For all mutations and polymorphisms, the absolute no and % should be included.

Author Response

Dear Reviewer,

We would like to thank you for yours suggestions/comments which will improve the quality of this article if accepted. We have answered all questions point by point and included the recommendations in the document; the changes are highlighted in yellow. Please find attached the revised documents (reposponses to Reviewer and main document).

Sincerely,

Reviewer 2 Report

In the manuscript “Prevalence of HIV-1 Natural Polymorphisms and Integrase Resistance Associated Mutations in African Children”, the authors described a molecular epidemiology survey of HIV integrase resistance mutations and non-resistance polymorphisms in children with or without antiviral therapy in two western African countries. The paper is generally well written and provided relevant information to clinicians and researches in the field.

Here are my comments and suggestions

1. The two groups of children were recruited from Benin and Mali. How different is the HIV-1 epidemic in these two countries? Are the circulating major HIV-1 variants comparable in these two countries?

2. It seems like the children with treatment failure were less likely to be amplified.

3. Line 105-106. Here the authors listed three groups of children, treatment failure, ART-treated, and ART-naïve, similar as Figure 1 and Figure 2. But in the previous section, the authors only listed two groups of children. ART-failure and naïve. Please clarify.

4. Table 1 can be improved in terms of the format. Here is my suggestion. It is also important to list some of the characteristics for both ART-treated and ART-naïve groups.

Characteristics

ART-naïve (n = 44)

ART-treated (n = 31)

Total

Gender

    Male

35 (46%)

Median Age (Years and IRQ)

2.6 (1.5 – 5.0)

10.0 (6.0 -13.0)

CD4….

5. Please be consistent using “Figure” or “Fig.”

6. Describe the statistically analysis they did.

Author Response

Dear Reviewer,
We would like to thank you for yours suggestions/comments which will improve the quality of this article if accepted. We have answered all questions point by point and included the recommendations in the document; the changes are highlighted in yellow. Please find attached the revised documents (responses to Reviewer and main document).
Sincerely,

Reviewer 3 Report

This research investigates the presence of integrase resistance mutations in children with HIV living in Benin and Mali, West Africa. The paper is well written and the Methods and Results were presented clearly. Below are some inquiries/suggestions to improve the manuscript:

1. Please comment on the use of dried blood spots (DBS) and whether the sampling could have interfered with sensitivity in the detection of drug resistance mutations.

2. Do we have information on the mothers' genotype testing? Specifically, whether the mothers had virological failure or known resistance to integrase inhibitors.

3. The study included ART-naive children with median age of 2.6 years old. Although it may not be the main focus of the study, it would be helpful for the readers to understand the antiretroviral treatment guidelines of Mali and Benin, as current guidelines typically call for the immediate treatment of patients with HIV. Is there a reason why the patients are not on antiretrovirals?

4. Please provide a brief comment on the timing of resistance testing in the current cohort and the detection of archived viruses. It is worthy to note that transmitted drug-resistant variants are frequently outcompeted by wild-type variants among patients newly diagnosed with HIV.

Author Response

(The authors gave the same response as above.)

Round 2

Reviewer 1 Report

Thanks for the revision it improves the quality of the manuscript.